# Microscale Modeling of Frozen Particle Fluid Systems with a Bonded-Particle Model Method

**DOI:** 10.3390/ma15238505

**Published:** 2022-11-29

**Authors:** Tsz Tung Chan, Stefan Heinrich, Jürgen Grabe, Maksym Dosta

**Affiliations:** 1Institute of Solids Process Engineering and Particle Technology, Hamburg University of Technology (TUHH), 21073 Hamburg, Germany; 2Institute of Geotechnical Engineering and Construction Management, Hamburg University of Technology (TUHH), 21073 Hamburg, Germany

**Keywords:** discrete element method, bonded-particle model, frozen particle fluid systems, material modeling, material micromechanics, creep

## Abstract

An inventive microscale simulation approach is applied to investigate the mechanics of frozen particle fluid systems (PFS). The simulation is based on the discrete element method (DEM) and bonded-particle model (BPM) approach. Discrete particles connected by solid bonds represent frozen agglomerates. Uniaxial compression experiments were performed to gather data for material modeling and further simulation model validation. Different typical mechanical behavior (brittle, ductile, dilatant) were reviewed regarding strain rates, saturation levels, and particle mechanical or surface properties. Among all these factors, strain rate significantly affects the mechanical behavior and properties of the agglomerates. A new solid bond model considering strain-dependent and time-dependent behavior is developed for describing the rheology of the frozen particle fluid systems. Without alternating Young’s modulus and Poisson’s ratio of the bond material, the developed solid model provides a suitable agreement with the experimental results regarding different strain rates.

## 1. Introduction

Frozen particle fluid systems (PFS), such as frozen agglomerates or frozen soils (grounds), are classified as composite materials that have been vastly investigated in academic and industrial fields. Regarding natural science, frozen soils have been studied for centuries. Due to building projects being developed closer to the arctic region, artificial ground freezing (AGF) was introduced as a temporary stabilizing technique for mining and construction projects [1]. In the technical particle field, the exploration of the interplay of granular and liquid phases at freezing temperature, as well as the resulting micromechanical behavior of the final composite, plays an essential role in many processes, starting from the storing of humid materials at low temperatures in the silo, ending with cryogenic grinding of temperature-sensitive materials.

A considerable portion of frozen PFS is ice, which has been studied in the academic field for centuries. Mechanics of ice were well investigated throughout different scopes, such as the mechanical properties of polycrystalline ice and columnar-grained ice [2,3], viscoelastic properties [4], creep behavior [5], temperature effects on creep behavior [6], the influence of surface properties on bond interface performance [7], critical factor influence the formation of ice bond [8]. Recent research has attempted to develop an artificial neural network to establish a predictive database for ice mechanical behavior regarding ice type, temperature, and strain rate [9].

Nonetheless, simulation tools were applied to investigate the mechanical behavior of ice. For instance, the finite element model (FEM) was used for the high dynamic behavior of ice [10], the interaction between ship structure with broken ice by the elastic ice model [11], or the discrete element method (DEM) used for interaction between a conical structure with sea ice in the arctic region [12].

Apart from the bond material alone, the composite of frozen PFS has been investigated for centuries due to different applications related to frozen soils. Available literature has investigated rheology on frozen soil and the application of AGF [13,14,15].

Similarly, numerous simulation works have been carried out on frozen soil and particle-reinforced composite failure progress with FEM and xFEM (extended finite element method) [15,16]. Resolving crack propagation with xFEM showed promising results. However, re-meshing was required after every consecutive step due to the changing geometry. FEM coupling with a thermo-hydro-mechanical model has been conducted to tackle the interaction between thermal, hydraulic, and mechanical loads [17]. Similar works using DEM to resolve the mechanical problems for frozen soils have been carried out [18,19], with creditable results revealed from the simulation.

The particle-based mesh-free discrete element method (DEM) is a numerical model for understanding particle dynamics introduced by Cundall and Strack [20]. The bonded-particle model (BPM) extends the soft-sphere formulation of DEM [21], in which solid bonds are created to connect primary particles and form the agglomerates. During the simulation, each bond is treated as a separate entity and can be removed or created to mimic fracture or even material sintering. Both BPM and DEM have been applied for tackling different mechanical problems, including damage progress of concrete or high-performance concrete [22,23], cemented sand [24], rock mechanics [25], or mechanics of biopolymer aerogel [26], and many other materials. The main advantages of BPM simulation are:Flexibility in agglomerate generation, in which all particles and bonds can have their unique material or geometrical properties;Capability in mimicking the breakage behavior of agglomerate, such as the crack initiation, propagation, failure plane, etc.;Diversity in functional model usage, with numerous choices of rheological models in the particle-particle, particle-wall relationship, and solid bond models.

The application of BPM demands high computational power due to the massive number of objects considered in the simulation and the small simulation time step. However, different parallelization techniques, especially focused on applying graphic process units (GPU), have efficiently compensated for such deficiencies [27].

In this contribution, a new solid bond model that combines strain-dependent linear elastic behavior with time-dependent creep behavior has been developed and integrated into the open-source DEM framework (MUSEN) [27]. The uniaxial compression experiment has proceeded for material parameter calibration and simulation model validation. Detail of the solid bond model and comparison between experimental and simulation results are discussed.

### 1.1. Ice Rheology

Ice rheology has been investigated for decades, with different literature analyzing the mechanical properties, viscoelastic properties, and creep behavior. Young’s modulus of ice ranges between 9.7 and 11.2 GPa and Poisson’s ratio from 0.29 to 0.32, obtained by the biaxial bending of ice plates at approximately −10 °C [28,29]. Ice tensile strength and compression strength react differently concerning temperature and strain rate. Ice’s tensile strength ranges from 0.7 to 3.1 MPa, and compressive strength ranges from 5 to 25 MPa. The temperature-weakening effect on tensile strength is less than compressive strength [30].

Compared to temperature, strain rate has almost no effect on tensile strength but vastly alternates the compressive strength of ice. From 10^−8^ s^−1^ to 10^−3^ s^−1^, the compressive strength increases with an increase in strain rate, surpassing 10^−3^ s^−1^ compressive strength decreases with an increase in strain rate [29].

Apart from mechanical properties, mechanical behavior is crucial to be identified. Ductile, dilatant, and brittle behavior can be identified in ice, mainly characterized by strain rate. In the case of high strain rates, brittle behavior prevails. Under tensile stress with a high strain rate, the resistance to ice damage can be described by the nucleation and growth of cracks. The strength is limited by the grain size of the ice, which can vastly alternate the crack propagation. Under compressive strain, abrupt collapses occur at around 0.5% strain, in which the shear plane is located around 30° to the maximum principal stress plane [31].

Ductile behavior is mainly identified by strain-rate hardening and thermal softening, in which the activation energy almost doubled above −10 °C. The dislocation-based process dominates the primary behavior under low strain rate deformation. The deformation relationship is quantified by quasi-static creep [32], where the material experience permanent deformation under stress, which is well below the yield stress for a prolonged duration. Three different creep phases can be identified with decreasing, steady, and increasing strain rates [33]. The power law can be used to formulate a rheological model for the creep behavior of ice [5]. From this numerical relationship, the main dominance of ice under low strain rate loading depends on applied stress and temperature [31]. The power law for creep (Equation (1)) describes the creep strain rate ${\dot{\varepsilon }}_{n,cr}$, which depends on the applied stress σ and two model parameters *A* and *m*. These parameters should be adjusted to considered different material and temperatures:(1)$${\dot{\varepsilon }}_{n,cr}=A\;\cdot \;\sigma^{m}\;$$

### 1.2. Rheology of Frozen Soil

Frozen soil shares remarkable similarities with ice in mechanical behavior, which is closely related to frozen PFS. Suitable insight into frozen PFS can be given by interpreting the characteristic of frozen soil. It has four typical mechanical behaviors: brittle failure, brittle behavior with failure just after the yield point, ductile behavior with strain hardening, and strain softening [34]. Literature regarding permafrost soil samples and artificially frozen soil has confirmed such behavior [35,36]. Apart from the strain rate, volumetric ice content plays a vital role in frozen soil’s mechanical behavior. The ice content influences the frozen soil’s ductility or brittleness [37]. Furthermore, the temperature, salinity, dynamic load, and refreezing slightly influence the mechanical strength and behavior of the frozen soil [38,39,40,41,42].

Regarding the unfrozen water in the frozen soil, various works of the literature showed a remarkable effect on the frozen soil’s mechanical and creep behavior [43,44]. Due to the incompressible nature of water, unfrozen water in the frozen soil can transfer both negative and positive pressure. Still, water can be discharged with drainage, and the frozen soil’s original form of stress state can be reestablished. However, due to its complexity, unfrozen water is not considered in both the experimental or simulation stages.

## 2. Materials and Methods

### 2.1. Uniaxial Compression Test

A universal texture measurement system, T.A. X.T. plus Texture Analyzer (Stable Micro System Ltd., Surrey, United Kingdom), was used to perform uniaxial compression tests. This setup was coupled with a self-constructed climate chamber, which aimed to maintain the ambient temperature of the inner cavity under the freezing point, thus preventing any thermal failure of specimens. It consisted of 3D-printed parts, coupled with radiators and radial fans, connected with a cryogenic unit (IKA Temperature Control, RC2 basic, lowest temperature: −20 °C). The chamber could maintain the ambient temperature at −10 °C, with less than 0.3 °C deviations. This experimental setup required a metal punch and base to protect the load cell and Texture Analyzer base, which were cooled down passively by the cool air in the climate chamber. Additionally, a 3D-printed fragment container is created to capture any fragment generated during the experiment, preventing the ice-particle pieces from being trapped in the radial fan. Three-dimensional CAD drawings of the climate chamber and coupled configuration are shown in Figure 1.

### 2.2. Specimen Preparation

The experiments have been carried out for pure ice samples and afterward for the frozen PFS. Here, accurate specimen preparation has played a crucial role in the credibility of the experiment data since undesired defects influence the data’s accuracy. For this purpose, special molds constructed with silicon have been used for the ice and PFS freezing process, as silicon is excellent for specimen extraction.

Nevertheless, different challenges have been faced during the preparation of samples. For example, freezing water in silicon mold was not acceptable. Defects, such as bulging, have been observed. Bulging was caused due to water expansion, as water decreases in density during the phase transition stage. The phase transition occurs from outside to inwards, forming a rigid shell of frozen water surrounding the inner unfrozen part. As the inner part expands during further freezing, such expansion breaks the outer shell in different directions. It creates different bulging or defects, leading to the sample’s unpredictable geometry.

Hence, polycrystalline ice [6] was produced for the experiment based on the method proposed by [45], formed by the compression of snow in the initial stages. Firstly, crushed ice (1.12 mm–1.7 mm) formed after the freezing of distilled water was packed into the mold, and 0 °C distilled water was then injected from the bottom. Then, the specimen froze in the household refrigerator under −18 °C. The ice grain inside is randomly oriented to create a homogeneous structure. In addition, ice specimens produced by such a method were mainly formed from pre-frozen ice. The expansion rate of the specimen was kept under control, which allowed more delicate monitoring of stress and strain calculation.

Different particle types were considered for the frozen PFS specimen to cover the broadest range of interest. Samples of 10 ± 0.05 mm and 8 ± 0.05 mm in diameter were created, with a height-to-diameter ratio ranging around 1.6 ± 0.1. The 8 mm mold was designated to achieve higher stress, as the Texture Analyzer load cell can only withstand 500 N. The maximum compressive pressure achieved with the setup is 6.37 MPa (10 mm diameter) and 9.95 MPa (8 mm diameter), respectively.

An overview of primary particles utilized for sample preparations and their classification is provided in Table 1. Agglomerates are constructed by primary natural (obtained from nature without any shape or surface alternating processing) or technical (manufactured according to predefined geometry and surface properties) particles.

The saturation level of investigated PFS was either 100% or 75%. The definition of saturation level is calculated according to the remaining volume, which is not occupied by particles. The number of particles in the specimen is the same across different saturation levels. The saturation level governed by what portion of the remaining volume was occupied by bond material is calculated as follows:(2)$$Saturation\;level=\frac{Volume_{bond}}{Volume_{specimen}-Volume_{particle}}\cdot 100\%$$

The samples with 100% saturation were prepared as follows. Firstly, primary particles were poured into the mold. Afterward, deionized water was injected from the mold’s bottom and passed to the degassing chamber to avoid forming gas bubbles; hence no undesired defects existed in the agglomerates. For 75% saturation samples, a predefined amount of water was sprayed onto the particles, and then the particles were thoroughly mixed and filled into the mold.

Both types of PFS were then frozen overnight in a household refrigerator at −18 °C. PFS specimens were extracted from the silicon molds, and necessary adjustment with a utility knife blade was applied if the contact surface was not perpendicular to the sidewall. Two final samples with a diameter of 10 mm are shown in Figure 2.

### 2.3. Investigated Parameter Space

Since the strain rate has a decisive influence on the rheological behavior of ice and frozen soil [1,31], two different strain rates have been tested to review different PFS mechanical behavior. A total of 10^−3^ s^−1^ was applied for a low strain rate, and 10^−2^ s^−1^ was used for a high strain rate. The strain rate is controlled by the compression speed, specified according to specimen height. The compression speed of 0.02 mm/s, which is two times larger than the minimal compression speed of the Texture Analyzer, was chosen as the low strain rate for all specimens. For high strain rate, compression speed is calculated by specimen height×0.01. All experiments have been performed at a temperature of around −10 °C. The possibility of unfrozen water inside the agglomerate can be minimized with such a temperature.

### 2.4. Ice Creep Behavior

The polycrystalline ice specimen was loaded to a particular force, and the applied pressure was held for 240 s to review the creep behavior. Figure 3 shows the transition from the primary creep (decreasing in strain rate) to the secondary creep (steady in strain rate) in the experiment. However, an utterly constant strain rate was not yet achieved, which is crucial for calibrating the creep parameters for the simulation. In the literature, a steady strain rate was achieved after 100 min of force holding [32], which was impossible from the current experimental setup.

Compressive Young’s modulus was estimated according to the primary loading part, in which a linear stress-strain relationship was observed. Several demonstration stress-strain curves are presented in Figure 4. Representative stress-strain curve of polycrystalline ice during creep experiment (primary loading phase) A linear relationship is fitted and the slope is obtained from the linear portion of the curve. The fitted curve is marked with a straight line on the figure. The compressive Young’s modulus estimation does not consider the primary loading portion. As this portion usually has a lower stiffness. It is related to the surface of the ice column is not entirely in touch with the punch, and the punch was deforming the surface rather than compressing the complete column of ice. The compressive Young’s modulus is 346.575 ± 48.2687 MPa.

### 2.5. Fracture Patterns of Frozen PFS

Figure 5 shows the breakage pattern of the frozen PFS of different saturation levels during uniaxial compression under a high strain rate (10^−2^ s^−1^). The fully saturated PFS samples can better transmit the pressure through the specimen. Thus, the cracks propagate from top to bottom, breaking the specimen into relatively large fragments containing numerous primary particles. In this case, the complete failure of the specimen occurred vigorously. In contrast, no large fragments were formed for PFS samples with 75% saturation. Only tiny pieces with several primary particles detached from the main structure during loading.

One additional phenomenon observed was the inhomogeneity of 75% saturation level PFS. The PFS is prepared by freezing particle water mixture. As the fridge temperature is kept at −18 °C, the water in the agglomerate cannot be frozen instantly. During the slow freezing process, part of the water around the particle performs a phase transition, and the remaining water is concentrated toward the lower portion of the agglomerate, forming an inhomogeneous specimen. Such a slow freezing process also imposes internal and bond structure differences concerning different primary particles. As the particle surface topography, particles’ shape, separation distance, and liquid bridge size influence the successfulness of liquid bridge formation [46,47,48]. Thus, only the successfully formed liquid bridge forms a solid bond. This causes the differences in internal structure and bond structure to differ concerning different types of primary particles. However, it is impossible to gather the difference in the bond structure and internal structure in 75% saturation level PFS between different primary particles with the current experimental setup.

### 2.6. Mechanical Behavior of Frozen PFS

All possible mechanical behaviors can be observed in the experiments performed under different conditions, such as varied types of primary particles, strain rate, and saturation levels.

Figure 6 shows the stress-strain curve of some representative experiments for PFS with sand and alpha-alumina primary particles under high strain (HS) and low strain (LS) rates. The results show that the sand PFS under high strain rate loading reveals brittle failure. In the case of alpha-alumina and high strain rates, brittle behavior with failure is observed just after the yield point. Finally, ductile behavior with strain hardening is typical for sand PFS under low strain rate loading and strain softening for alpha-alumina PFS under low strain rates.

### 2.7. Bonded-Particle Model Approach

Open-source GPU-accelerated DEM framework MUSEN was used to describe frozen PFS behavior. Among others, this system supports calculations with the bonded-particle model (BPM) [27]. In BPM, the spherical primary particles can be connected through solid bonds. Every single bond can have its unique dimensions and material properties. Bonds can be created or destroyed during simulation to mimic sintering or fracture behavior. The bond is modeled as a virtual cylindrical linkage between the particles. Bonds are treated as virtual objects with no volume or mass, and the internal force is calculated according to strain changes; in our case, time is also considered in the solid bond mathematical model.

Agglomerate is generated with two consecutive steps. The packing of primary particles is generated in a virtual volume according to the force-bias algorithm [26]. The virtual volume represents the geometry of the specimen. All particles are randomly generated in this pseudo space, which means no particles were generated outside the virtual volume. The number of particles is governed by the porosity preset in the particle generation progress. In each iteration, the overlaps between particles are detected, and force, which is proportional to the overlap, is calculated. Afterward, the primary particles are shifted according to this force. Generation completes if the maximum overlap is smaller than the target value. Finally, to build the agglomerate, particles are connected with solid bonds. The generation of bonds is governed by the minimum ($L_{gen}^{min}$) and maximum distance between the surface of spheres ($L_{gen}^{max}$). If the distance between the surfaces ranges between the preset value ($L_{gen}^{min}$ and $L_{gen}^{max}$), bond is created. As mentioned, the bonds are treated as virtual objects; thus, overlapping between bonds is allowed. By alternating the limiting value $L_{gen}^{max}$, different amounts of bonds can be generated inside the same particle packing. In most cases, minimum distance ($L_{gen}^{min}$) is set to a negative value, as particle overlapping is allowed during package generation.

### 2.8. Solid Bond Model Considering Creep Behavior

The newly developed solid bond model aims to tackle the strain rate-dependent behavior of the bond material. Therefore, the component to consider the creep behavior has been included in the model. The particle-particle and particle-wall interactions are calculated according to the Hertz–Mindlin model, whereby the normal force is calculated according to Hertzian theory [49], and the tangential force is calculated according to the model proposed by Mindlin et al. [50].

During simulation, the calculation of bond force is separated into the normal and tangential directions. The newly developed model coupled the strain-dependent elastic bond model with the time-dependent creep model. As schematically shown in Figure 6, the primary deformation stage of frozen PFS is in a linear elastic relationship [51]. The linear elastic strain-dependent relationship describes the primary loading phases of bonds. The total strain $\varepsilon_{n,to}$ in normal direction is calculated based on the initial $L_{I}$ and current bond length $L_{c}$:(3)$$\varepsilon_{n,to}=\frac{L_{c}-L_{I}}{L_{I}}$$

Total strain in the normal direction can be decomposed into two parts:(4)$$\varepsilon_{n,to}=\varepsilon_{n,el}+\varepsilon_{n,cr}$$
where $\varepsilon_{n,el}$ is the elastic strain and $\varepsilon_{n,cr}$ is the irreversible deformation due to creep. The simulation automatically replaces plastic deformation with creep deformation [52]. Plastic deformation is regarded as an inelastic deformation, effectively the deformation due to creep under viscoelastic conditions. Normal bond stress $\sigma_{n}$ is calculated by:(5)$$\sigma_{n}=\{\begin{array}{c} E\cdot \varepsilon_{n,el}\;\;if\;\sigma_{n}<\sigma_{n,y}\;\\ \sigma_{n,y}\;\;if\;\sigma_{n}\geq \;\sigma_{n,y}\; \end{array}\;$$
where *E* is Young’s modulus of the bond material and the $\sigma_{n,y}$ is the yield strength.

According to Norton [53], the power law describes the creep behavior of solid bonds (Equation (1)). The power law can provide approximately the same behavior concerning different applied stresses, which means the equation can provide an approximate same “shape” regardless of different applied stress [54]. Change of temperature during simulation is not considered for the model simplicity. The irreversible creep strain in the solid bonds is calculated iteratively in each time step as:(6)$$\varepsilon_{n,cr}\left(t+\Delta t\right)=\varepsilon_{n,cr}\left(t\right)+\Delta t\cdot A\cdot \;{\left(\sigma_{n}\left(t\right)\right)}^{m}$$ Δ*t* denotes the simulation time step. The resulting bond force in the normal direction $F_{n,b}$ calculated by:(7)$$F_{n,b}\;\left(t+\Delta t\right)=E\cdot A_{b}\cdot \left(\varepsilon_{n,to}-\varepsilon_{n,cr}\left(t+\Delta t\right)\right)$$
where $A_{b}$ is the bond’s cross-cut area. With strain applied to the bond, normal stress response increased according to an approximately linear relationship. The calculation of bond normal stress during the loading part combines the linear strain-dependent relationship with the not linear time-dependent relationship. Bond normal stress increases until yield strength and holds. If strain remains unchanged after a specific time, force response decreases gradually, calculated according to the creep parameter and the previous time step’s bond normal stress. A demonstration of the stress-time relationship for a single bond is shown in Figure 7.

Creep behavior is considered in both normal and tangential directions. The same power law of creep describes the creep behavior in the tangential direction. Unlike the calculation of strain in the normal direction, the bond deformation in the tangential direction ${\vec{\delta }}_{t}$ in every simulation step is updated according to the previous time step relative motion:(8)$${\vec{\delta }}_{t}\left(t+\Delta t\right)=T\cdot {\vec{\delta }}_{t}\left(t\right)+{\vec{v}}_{t,rel}\;\cdot \;\Delta t$$
where T is the rotation matrix to consider the motion of connected particles between the current and previous time step [55], and ${\vec{v}}_{t,rel}$ is the relative velocity at the contact point in the tangential direction.

The tangential stress, which is crucial for the calculation of the tangential creep strain rate, is calculated according to the following:(9)$$\sigma_{t}=\frac{\left|{\vec{\delta }}_{t}\right|}{L_{I}}\cdot \;\frac{E}{2\;\cdot \left(1+v\right)}$$
where v is the Poisson’s ratio of the bond material. The same power law with the same creep parameters A and m are used for the creep behavior in the tangential direction. Strain change due to creep in the tangential direction after every consecutive time step is calculated according to:(10)$${\vec{\varepsilon }}_{t}\left(t+\Delta t\right)={\vec{\varepsilon }}_{t}\;\left(t\right)+\varepsilon_{t,cr\;}\left(\Delta t\right)\;\cdot \;{\vec{r}}_{t}$$
where ${\vec{r}}_{t}$ is the unit vector of the bond in the tangential direction, which is defined by:(11)$${\vec{r}}_{t}=\frac{{\vec{\delta }}_{t}\left(t\right)}{\left|{\vec{\delta }}_{t}\left(t\right)\right|}$$

The resulting tangential force can be calculated by:(12)$${\vec{F}}_{t,b}=\frac{{\vec{\delta }}_{t}\left(t+\Delta t\right)}{L_{I}}\cdot \frac{E}{2\;\cdot \left(1+v\right)}\cdot A_{b}$$

Apart from the agglomerate mechanical behavior, the agglomerate fracture is considered in the model. This is accomplished by comparing the individual bond stresses under loading with material properties such as normal $\sigma_{max}$ and tangential strength $\tau_{max}$ as well as comparing the total strain and critical breakage strain $\varepsilon_{n,max}$ as:(13)$$\frac{{\vec{F}}_{n,b}}{A_{b}}+\frac{{\vec{M}}_{n,b}\cdot R_{b}}{I}\geq \sigma_{max}$$
(14)$$\frac{{\vec{F}}_{t,b}}{A_{b}}+\frac{{\vec{M}}_{t,b}\cdot R_{b}}{J}\geq \tau_{max}$$
(15)$$\varepsilon_{n,to}\geq \varepsilon_{n,max}$$
where $R_{b}$ is the bond radius, ${\vec{M}}_{n,b}$ and ${\vec{M}}_{t,b}$ are the bending and torsional moments of the bond, respectively, I is the moment of inertia, and J is the polar (torsional) moment of inertia of the bond’s cross-section.

Pressure melting and recreation of bonds are not considered in this solid bond model, as it vastly increases the complexity of the mathematical model, which has to consider the temperature, pressure, and factors governing the recreation of the bond.

The normal compressive strength alternated according to strain rate [29]. However, due to the nature of the model, implementing a relationship for calculating normal and tangential strength according to strain rate causes massive fluctuation in both values. The newly developed solid bond model has not considered the strain rate-dependent compressive strength relationship. Variance in bond normal strength due to strain rate or particle surface properties is compensated by alternating the bond normal and tangential strength concerning PFS constructed by different primary particles under different strain rates.

## 3. Result and Discussion

### 3.1. Experimental Result

The overview of Young’s modulus and breakage stresses of frozen PFS under different saturation levels are presented in Figure 8. A reduction in the proportion of the bond material in the agglomerates vastly weakens the mechanical structure. This phenomenon can be observed from the value of the average Young’s modulus and breakage stress, in which the reduction in saturation level decreases both values, incredibly massive in breakage stress. The reduction ratio in mechanical strength is significantly higher than the reduction ratio in the bond material volume fraction.

For 100% saturation level PFS samples, Young’s modulus increased with the increase in strain rate, except for the frozen PFS with glass particles. The strain rate of PFS with glass particles does not significantly influence Young’s modulus. One of the possible reasons for that is the different creep behavior of ice in the contact zone between particle and bond. From Table 1, the glass bead has the lowest value in both Ra and Rz, which means the glass bead has the smoothest surface among all the primary particles. The smooth surface affects the contact zone behavior. This phenomenon can also be observed for the breakage stress of glass PFS with a 100% saturation level, where the aggregate reveals the lowest breakage strength under both strain rates. Overall, for all investigated materials, it can be observed that with an increased roughness, the strength increases, and the highest strength reveal aggregates containing non-spherical sand particles.

Young’s modulus for sand PFS samples at different strain rates is also a vast difference. The surface properties of sand particles are similar to Polyethene particles. The much higher Young’s modulus at a high strain rate is related to the primary particle’s mechanical properties but is mainly associated with a lower porosity and, as a result, a more significant number of contacts for each particle. Consequently, a more compact and refined internal bond structure is formed inside such samples, which increases stiffness.

On the other hand, the lower value of Young’s modulus at a low strain rate might be related to the pressure melting phenomenon, when the ice melts under pressure and then freezes again due to particle realignment. Due to the massive number of bonds in the sand PFS and the viscoelasticity of ice, this process can escalate, which causes a significant drop in Young’s modulus at a low strain rate. However, the current experiment setup cannot prove such an assumption. The strain rate for saturation level 75% only slightly affects Young’s modulus. However, for an unambiguous conclusion, it is necessary to conduct more experiments here, especially crucial data regarding the internal structure under pressure.

The average breakage stress of 100% and 75% saturation level PFS provided additional information about the mechanical behavior, presented in Figure 8. At 100% saturation level, an increase in strain rate leads to a higher fracture strength, except for the glass PFS, since the glass particle reveals a much smoother surface than other particles. The smooth surface leads to a weak interface between the bond material and the particle surface. At 75% saturation level, the high deviation within the experiment data does not allow for observing a clear trend except for the sand PFS, where it can be concluded that strain rate causes an increase in breakage stress. Apart from quantitative analysis of mechanical behavior, which is based on breakage stress, strain, and Young’s modulus, the overall rheological behavior should be considered. Table 2 gives an insight into agglomerate behavior for different saturation levels, temperatures, strain rates, and particle properties.

Particle surface properties and strain rate can influence mechanical behavior. Furthermore, the 75% saturation samples yielded more stepwise breakage under loading.

### 3.2. Simulation Setup

The agglomerates with 10 mm in diameter and 16 mm in height have been generated according to the section “Bonded-Particle Model Approach” procedure. The diameter of primary particles was tailored to the particles used in the experiment. Additionally, the driving velocity of the upper moving geometry (metal punch) was increased to 200 times (low strain rate: 4 mm/s, high strain rate: 32 mm/s) compared to the experiments to reduce the total computation time. The increase in moving geometry speed can lead to emerging of artificial elastic waves propagating through the material. The restitution coefficient for all particles and particle-wall interactions was reduced to 0.1 to prevent such a phenomenon. The restitution coefficient controls the amount of kinetic energy dissipated during simulation. In addition, the creep parameter has been adjusted accordingly.

For the spherical particles (polyethene, glass, alpha-alumina), spheres were used to represent the particles in the simulation. For non-spherical sand particles, the simulation has also used spheres to reproduce the particles due to contact detection occupying significant calculation power in DEM simulation. Computing the contact for spheres is less demanding, determined by the center distance being more or less than the distance of the combined radius. With non-spherical particles, not only is the distance considered, but also the relative rotation of particles needs to be taken into account. The simulation of non-spherical particles in DEM is tremendously more demanding than spherical particles [56].

The main model parameters and material properties of primary particles are listed in Table 3. The polymer particles, glass beads, and alpha-alumina Young’s modulus were determined from self-performed experiments. Here the force-displacement characteristics obtained from uniaxial compression tests were used to adjust Young’s modulus by fitting the Hertz model to experimental results [57]. Additionally, due to the non-spherical nature of sand particles, a uniaxial compression test for material modeling parameter calibration is impossible. In contrast, Young’s modulus of sand particles was taken from the literature [58]. Both density and Poisson’s ratio of different particles were taken from the literature [59,60,61,62].

The particle-particle, particle-wall sliding, and rolling friction were referenced from different literature, with further fine adjustment per trial-and-error procedure [62,63,64]. According to Gu et al. [65], the non-spherical PFS particle-particle and particle-wall sliding and rolling friction coefficients have been adjusted. In the DEM simulations, the shape of particles was neglected, and all investigated particles were modeled as spheres. Due to such a simplified representation, the value of rolling friction for non-spherical sand particles was much higher compared to spherical particles.

The material parameters for ice bonds have been adjusted according to experiments with 100% saturation. The simulation results are compared to averaged experiment stress-strain curve.

Young’s modulus maintained the same for all different sets of experiments and was estimated at 350 MPa. The Poisson’s ratio was taken from the literature as 0.31 [29]. Only the normal strength, tangential strength, and creep parameters *A* and *m* were tuned to individual experimental sets. The adjusted parameters for different sets of experiments are listed in Table 4. Particles investigated in this study have been selected due to their significant mechanical and surface properties differences. The creep parameters are adjusted by a trial-and-error procedure. It is related to two reasons. First, the creep parameter in the works of literature is calibrated concerning pure ice samples. The value cannot be applied to the simulation, as the contact zone influences the creep parameter concerning different primary particles. The second is the increased speed of moving geometry. Since the creep parameter is a time-dependent material parameter, any adjustment toward the simulation scene related to time influences the creep parameter setup.

Similar to the creep parameters, bond normal and tangential strength are tuned by the trial-and-error procedure. Consequently, the contact zone between ice bridges and particles was significantly varied. It is expected that the strength of the contact zone for smooth glass particles was much smaller than the similar strength for rough and highly porous alpha-alumina particles. Therefore, the bonds’ normal and tangential strengths were varied to consider that effect. Primary sets of simulations were performed during the trial-and-error material parameter tuning procedure. Young’s modulus and Poisson’s ratio have kept unchanged throughout the process. Simulation results were transferred into the stress-strain curves, which were compared with the experimental result. The material parameters were then narrowed down to achieve accurate simulation results compared to the experiment result. Creep parameters concerning different primary particles are compared to validate the model capability. As shown in Table 4, only alpha-alumina PFS has a different creep parameter, which is correlated to its aggregated surface topography.

For modeling agglomerates with lower saturation degrees, the same material parameters for solid bonds were taken in Table 4. However, the bond generation parameters were alternated. New diameter and new maximal generation distance between particles $L_{gen}^{max}$ were specified.

Compared to the 100% saturation degree, for the bond generation in the case of agglomerates with a low saturation degree of 75%, the maximal distance $L_{gen}^{max}$ has been reduced to 0.01 mm. Such a value assumes that an ice bond is only formed when the particles are in contact or adjacent with a minimal distance. As mentioned, parameters constraining the successfulness of liquid bridge formation were related to the geometry, surface topography, separation distance, and size of the liquid bridge. As shown in Table 5, the total number of bonds for 75% saturation degree was almost halved compared to the 100% saturation level.

The cross-section area was also reduced to mimic the alternation in the internal bond structure. A quarter facilitates the bond diameter for agglomerates with spherical and non-spherical primary particles.

The assumption behind such a setup was the ice bridge formation deviation between particles in 100% and 75% saturation levels. At 100% saturation level, the cavity was occupied by water or particles, and both are incompressible substances. Solid bonds can completely enclose all primary particles inside the agglomerates. However, at a 75% saturation level, due to the volume reduction in bond substance, the individual geometry of the bonds between particles differs from the 100% saturation level agglomerate. The ice bridges are formed by the phase transition of the capillary liquid bridges. The starting geometry of ice bonds before phase transformation at 100% saturation level is entirely different from the 75% saturation level. Due to the difference in initial geometry, the deviation of the ice bond after phase transformation was accumulated. The ice bond of the 75% saturation is entirely different from the 100% saturation level. According to such an assumption, the reduction in bond diameter is applied to the simulation model.

In Figure 9, the agglomerates generated for different initial conditions are shown. The top part shows the agglomerates with spherical particles, and the lower part only shows the internal bond structure.

### 3.3. Comparison of Simulation and Experimental Results

The simulation and experimental results for the uniaxial compression test for different agglomerates with varied strain rates are shown in Figure 10. It can be seen that simulation provides acceptable values in breakage stress, strain, and Young’s modulus compared to experimental results. The proposed solid bond model can review different mechanical behavior regarding different strain rates.

Without the variation of bonds’ Young’s modulus and Poisson’s ratio, the proposed solid bond model can tackle the strain rate-dependent behavior with compensation of normal and tangential strength toward the particle surface properties. Last but not least, using alternated particle-particle and particle-wall friction coefficients to compensate for the shape deviation of simulation and experiment particles has proven feasible.

Apart from comparing the stress-strain data, the comparison between the experiment’s average Young’s modulus and breakage stress and simulation result is presented in Figure 11. All the simulation result lies within or very close to the average experiment value in both Young’s modulus and breakage stress.

The deviation between every experimental data was enormous in lower saturation level PFS. The experimental data and data obtained by simulation were compared by comparing the values of breakage stress and Young’s modulus.

The comparison of breakage stress and Young’s modulus of agglomerates with lower saturation degrees is shown in Figure 12. Simulations were not aimed at achieving the exact value, as the material parameter of particles and bonds for different conditions were not alternated. Only the internal bond structure is adjusted to compensate for the deviation of saturation level. Breakage stress obtained from the simulation yielded acceptable agreement with the experiment, as most of the simulation result lies within or close to the standard deviation range, except for the alpha-alumina PFS. The internal structure of the lower saturation level is generated according to arbitrary values, leading to a big difference compared to the realistic structure. The simulation and experiment comparison shows that the weakening effect of reducing saturation level differs remarkably toward alpha-alumina particles.

Young’s modulus from simulation reveals more significant deviations to experimental data than breakage stress. The main reason is the lack of detailed information on the contact zone between the bond and the particle. Such information is particularly essential for the particle-bond interface creep behavior, thus alternating the elastic behavior of the agglomerate.

Contrary to the assumptions made in the model, the internal structures of glass bead PFS and alpha-alumina PFS might not be the same. The capillary properties of glass and alpha-alumina are different, leading to the formation of varying bond networks. Furthermore, another critical role may play in the particle size and shape variability for the sand PFS. Thus, using the same settings for the bond generation applied for spherical and non-spherical PFS imposes out-of-range discrepancies in simulation and experimental results for non-spherical PFS.

## 4. Conclusions

Both experiment and simulation studies for frozen PFS were performed in this contribution. The experiments presented results that are in suitable agreement with previous literature. Primary particles with different surface and mechanical properties were constructed to validate the influence on the agglomerate mechanical properties and behavior. The saturation level was alternated during the experiment phase, which reviews a vast weakening effect on the agglomerate.

The bonded-particle model, an extension of DEM, has been used to tackle the problem. A new solid bond model considering creep behavior has been developed and implemented into the MUSEN simulation framework. The developed model can simulate strain rate-dependent material, formulating the different mechanical responses under different strain rates. Nonetheless, particle composite material experiencing creep behavior or damage can also be simulated.

Throughout all different simulations, the bond Young’s modulus and Poisson’s ratio have kept the same, with creep parameters, normal and tangential strength tuned according to primary particles and strain rate. The simulation model has proven to be capable of considering the strain rate-dependent behavior of the frozen PFS. In the lower saturation levels of frozen PFS, the bond’s material parameters were kept the same, and only the internal structure of the agglomerate in the simulation was changed. That shows a more considerable deviation in Young’s modulus but acceptable values in breakage stress. Lack of data regarding the internal structure leads to a higher variation of simulation results under a lower saturation level in PFS experiments.

## Figures and Tables

**Figure 1 materials-15-08505-f001:**
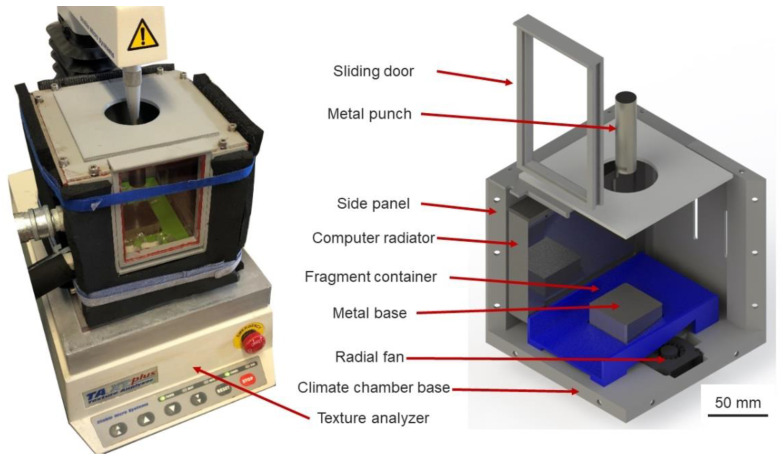
Texture Analyzer system equipped with a climate chamber: (**Left**) Climate chamber coupled with Texture Analyzer; (**Right**) CAD design.

**Figure 2 materials-15-08505-f002:**
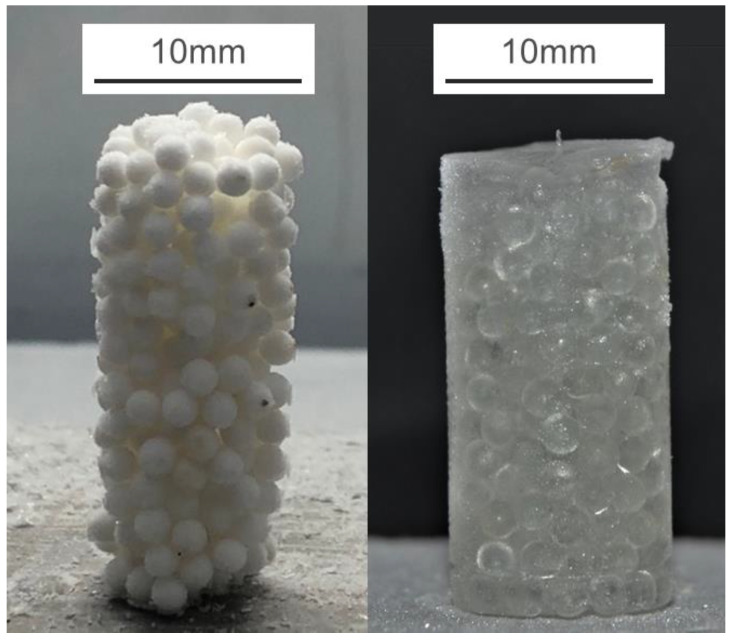
Two exemplary samples of frozen PFS: (**Left**) aggregate with alpha-alumina primary particles at 75% saturation level; (**Right**): glass bead PFS with 100% saturation.

**Figure 3 materials-15-08505-f003:**
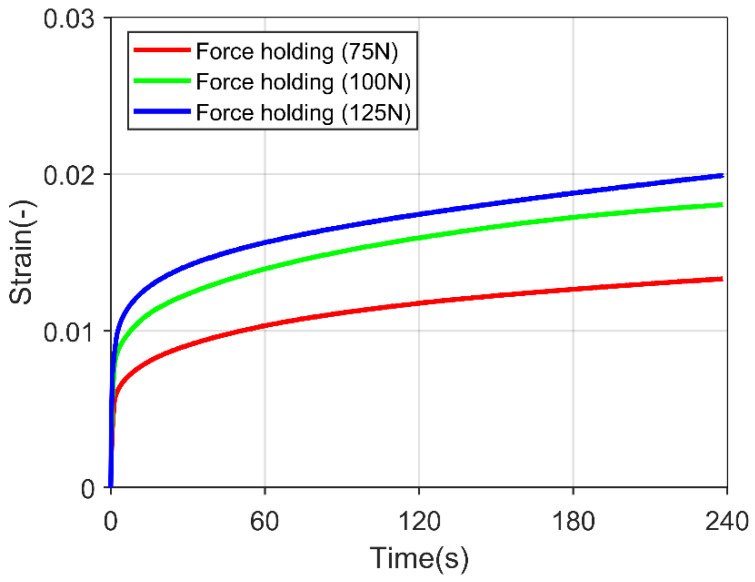
Polycrystalline ice specimen subjected to constant force load for 240 s under three different forces.

**Figure 4 materials-15-08505-f004:**
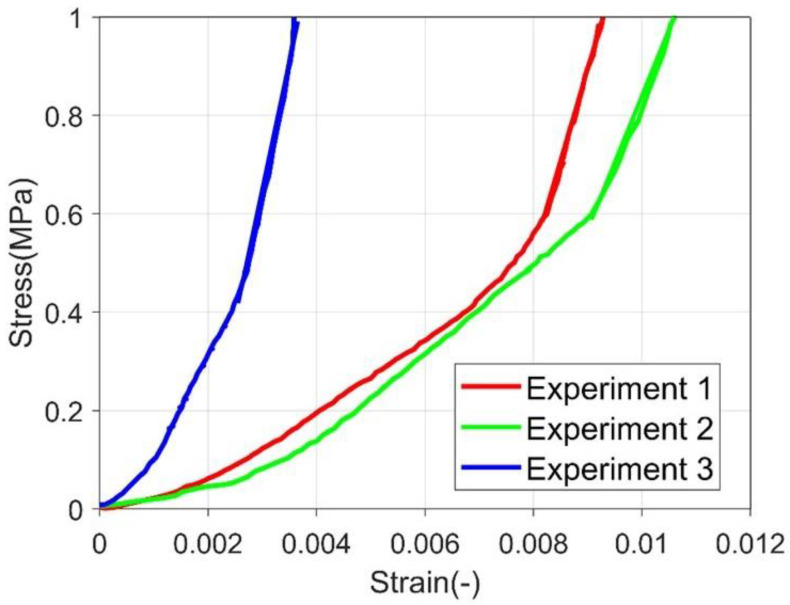
Representative stress-strain curve of polycrystalline ice during creep experiment (primary loading phase).

**Figure 5 materials-15-08505-f005:**
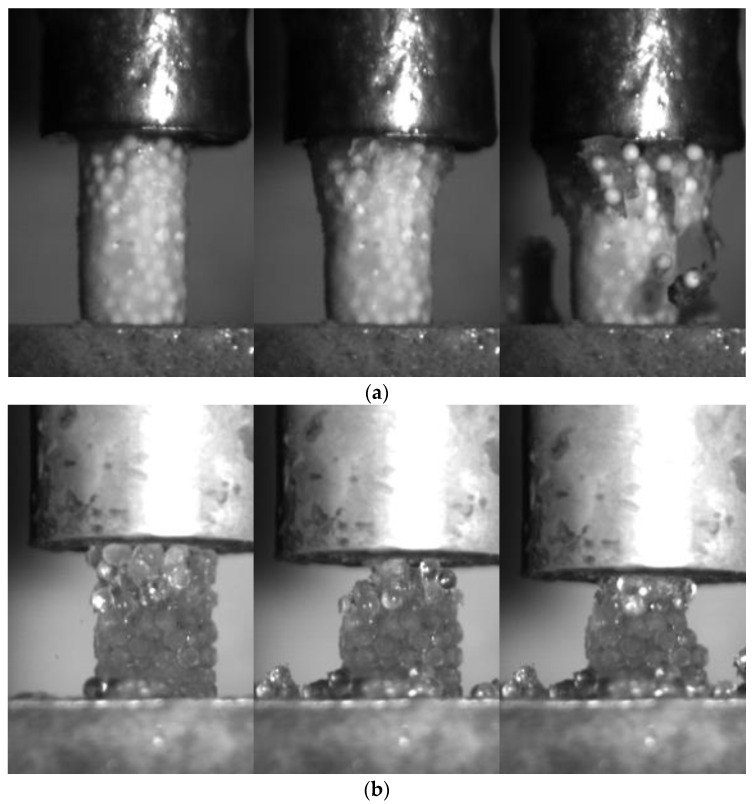
The fracture pattern for agglomerates at the strain rate is 10^−2^ s^−1^: (**a**) 100% saturation, alpha-alumina PFS (Appendix A); (**b**) 75% saturation, glass PFS (Appendix A).

**Figure 6 materials-15-08505-f006:**
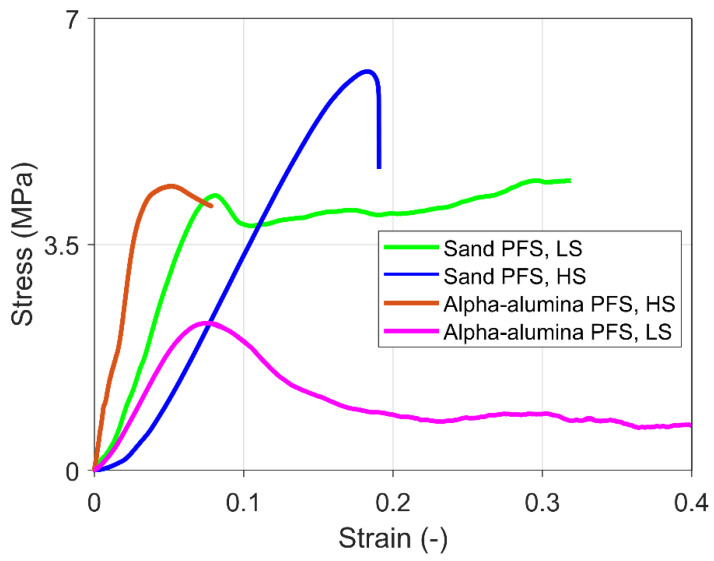
Representative stress-strain characteristics for different frozen PFS under a high strain (HS) rate of 10^−2^ s^−1^ and low strain (LS) rate of 10^−3^ s^−1^.

**Figure 7 materials-15-08505-f007:**
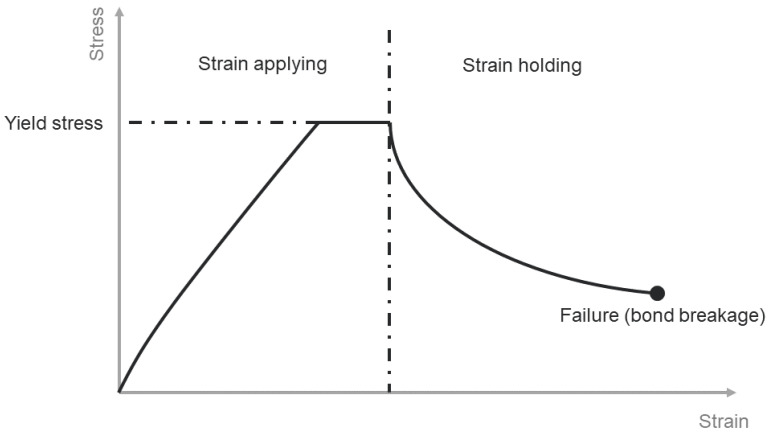
Schematic stress-strain diagram for solid bond under load in the normal direction.

**Figure 8 materials-15-08505-f008:**
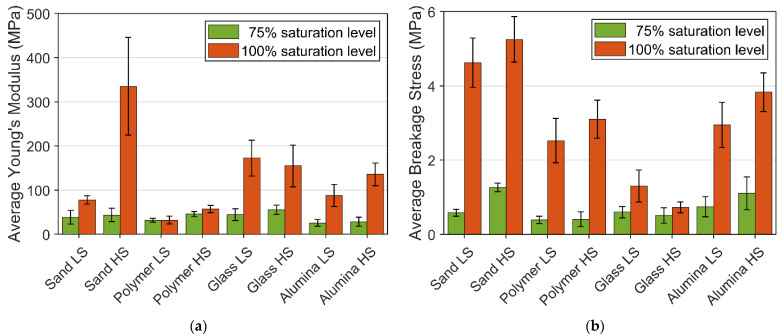
Average Young’s modulus and breakage stress of different PFS: (**a**) Young’s modulus; (**b**) Breakage stress.

**Figure 9 materials-15-08505-f009:**
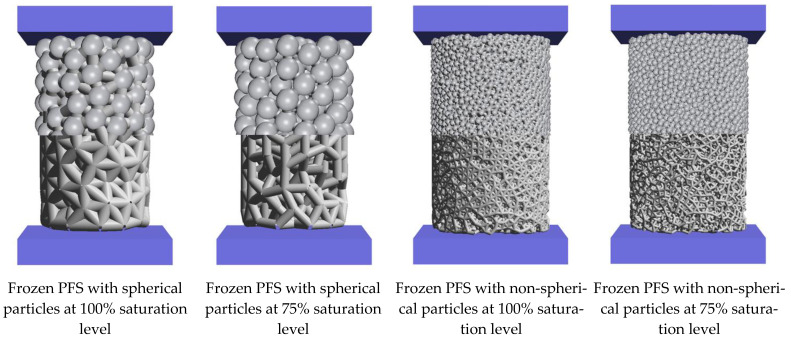
Representative agglomerate with diameter 10 mm, height 16 mm (**upper** part: agglomerates in complete form; **lower** part: agglomerates’ internal structure).

**Figure 10 materials-15-08505-f010:**
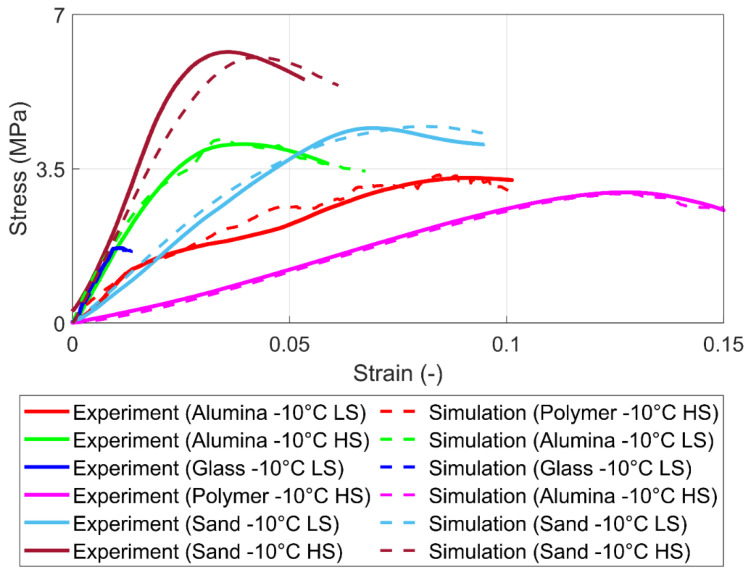
Experimental and simulation results for 100% saturation level for high strain (HS) rates (10^−2^ s^−1^) and low strain (LS) rates (10^−3^ s^−1^).

**Figure 11 materials-15-08505-f011:**
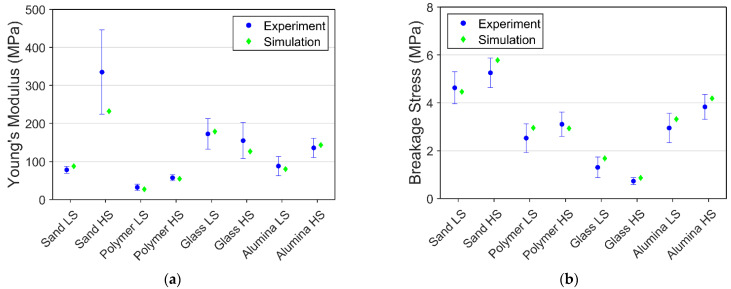
Comparison of experimental and simulation results for 100% saturation at −10 °C: (**a**) Young’s modulus; (**b**) Breakage stress.

**Figure 12 materials-15-08505-f012:**
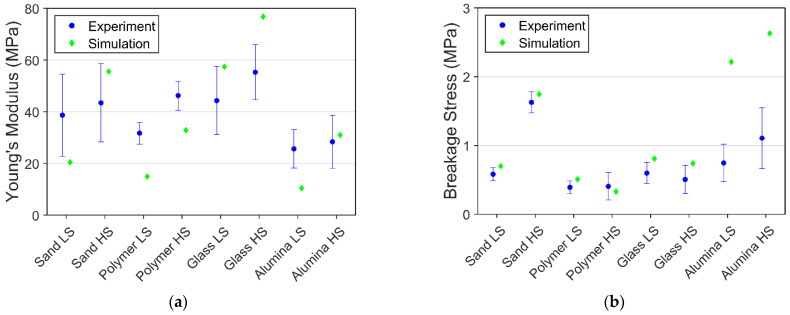
Comparison of experimental and simulation results for 75% saturation at −10 °C: (**a**) Young’s modulus; (**b**) Breakage stress.

**Table 1 materials-15-08505-t001:** Classification of primary particles used in experiments according to their properties.

	Stiffness		Shape		Surface Roughness		Particle Size (mm)
	Soft	Hard	Spherical	Non-Spherical	Ra	Rz	
Polyethene	X		X		12.808	50.723	1.8
Glass bead		X	X		1.767	11.462	1.65
Alpha-alumina		X	X		49.262	187.453	1.72
Quartz sand		X		X	13.416	49.623	0.5

**Table 2 materials-15-08505-t002:** Mechanical behavior overview concerning saturation levels, temperatures, strain rates, and particle properties.

Saturation Level	Strain Rate	Smooth Particles (Polymer and Glass)	Rough Particles (Sand and Alpha-Alumina)
100%	Low	Mostly brittle with failure just after the yield point	Dilatant with slight strain softening or hardening
	High	Brittle failure	Brittle behavior with failure just after yield or brittle failure
75%	Low	Brittle failure	Dilatant with vast strain softening
	High	Brittle failure	Brittle failure

**Table 3 materials-15-08505-t003:** Main agglomerates properties used for simulation of different types of PFS with 100% saturation level.

Parameter	Polyethene/Glass/ Alpha-Alumina (Spherical)	Sand (Non-Spherical)
Particle diameter (mm)	1.8/1.7/1.65	0.5
Bond diameter (mm)	1.0	0.3
Particle density (kg/m^3^)	960/2500/3960	2640
Particle Young’s modulus (GPa)	0.8/72.3/150	72
Particle Poisson’s ratio (-)	0.36/0.22/0.22	0.2
$\mathrm{Maximal}\;\mathrm{bond}\;\mathrm{generation}\;\mathrm{distance}\;L_{gen}^{max}$ (mm)	0.7	0.2
Numbers of particles (-)	≈230	≈11,200
No. of bonds (-)	≈1100	≈66,000
Porosity (-)	0.44	0.42
Particle-wall sliding friction (-)	0.45/0.45/0.45	0.45
Particle-wall rolling friction (-)	0.05/0.05/0.05	0.5
Particle-particle sliding friction (-)	0.45/0.4/0.45	0.45
Particle-particle rolling friction (-)	0.05/0.05/0.05	0.5
Restitution coefficient (-)	0.1	0.1

**Table 4 materials-15-08505-t004:** Material properties for ice bonds used for modeling different agglomerates at different loading rates.

	Primary Particles			
	Polyethene	Glass	Alpha-Alumina	Natural Sand
Normal and shear strengths				
- High strain rate (MPa)	3.5	4.2	20	20
- Low strain rate (MPa)	6	2.7	20	20
Creep parameter A (-)	0.1	0.1	0.3	0.1
Creep factor m (-)	0.1	0.1	0.16	0.1

**Table 5 materials-15-08505-t005:** Agglomerates bond generation set up for different types of PFS in 75% saturation level.

Parameter	Polyethylene/Glass/Alpha-Alumina (Spherical)	Sand (Non-Spherical)
Bond diameter (mm)	0.75	0.22
$\mathrm{Maximal}\;\mathrm{bond}\;\mathrm{generation}\;\mathrm{distance}\;L_{gen}^{max}$ (mm)	0.01	0.01
Number of particles	≈230	≈11,200
Number of bonds	≈550	≈34,000

## Data Availability

Data available on request.

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
