# Peer review of "Microscale Modeling of Frozen Particle Fluid Systems with a Bonded-Particle Model Method"

_materials, 2022, doi:10.3390/ma15238505_

Round 1

Reviewer 1 Report

This is an interesting paper that worth’s publication. However, the authors should perform several amendments to the manuscript before publication.

1)      Is there any effect of the size of the particles?

2)      The type of saturation also has an effect since in the 100% case the particles find themselves in an effective medium while in the case of 75% there is a kind of empty space that makes movement easier. Of course in the case of sand where the particle size is very small things are different. The authors should comment more on this issue

3)      In fig 4 it seems that as the 100% saturation sample is compressed it behaves more than a continuum with an increase of the diameter while the 75% sample breaks partly. This is also a point that worth’s further discussion. Could the authors perhaps attempt an effective medium approach.

It is apparent that there are several mechanisms that enter the phenomenon at various scales thus a multiscale approach could be foreseen (the authors could refer for example to

a)       Multiscale modeling in nanomaterials science. Materials Science and Engineering: C, 27(5-8), 1082-1089 (2007), (2017).

b)      Validity of Cauchy–Born hypothesis in multi-scale modeling of plastic deformations. International Journal of Solids and Structures, 115, 224-247 (2017).

Where lower scale methods are employed in order to extract more exact properties to enter to larger scale models.

4)      How the bonds are modeled on the BPM model

5)      How the frozen water is models in the BPM model. How the % saturation is taken into account in this modeling

6)      The agreement between experimental and simulation results is better as the authors mention too in the case of 100% saturation. Is this perhaps due to the fact that in that case the continuum model approach is more valid since there is no empty space as in the case of 75% saturation? And the BPM model as a continuum model performs better.

7)      Just a suggestion for the authors for future research. If they posses a significant number of experimental values they could attempt a machine learning approach to predict the values of  the Young modulus as function of the various parameters that enter the calculations (methods range from non transparent ML methods like neural networks (see for example Prediction of composite microstructure stress-strain curves using convolutional neural networks. Materials & Design, 189, 108509 (2020.) to transparent methods like symbolic regression (see for example A combined clustering/symbolic regression framework for fluid property prediction. Physics of Fluids.(2022))

Author Response

Response to Reviewer 1 Comments

Point 1: Is there any effect of the size of the particles?

Response 1: The size effect of the same particle types has not been conducted in the experimental phase. Only the sand particle has a much smaller particle size than other technical particles. However, the agglomerate is closely replicated in the simulation phase, in which the simulation model proved creditable concerning different particle sizes, material and surface properties.

Point 2: The type of saturation also has an effect since in the 100% case the particles find themselves in an effective medium while in the case of 75% there is a kind of empty space that makes movement easier. Of course in the case of sand where the particle size is very small things are different. The authors should comment more on this issue

Response 2: The primary particles' surface topography, particle shape, separation distance and size of the liquid bridge govern the successfulness of the formation of the liquid bridge at 75% saturation level. The internal and bond structures differ concerning different types of primary particles. However, with the current experiment setup, the difference of bond structure and internal structure at 75% saturation level concerning different primary particles cannot be revealed.

An additional description has been added to the article.

Point 3: In fig 4 it seems that as the 100% saturation sample is compressed it behaves more than a continuum with an increase of the diameter while the 75% sample breaks partly. This is also a point that worth's further discussion. Could the authors perhaps attempt an effective medium approach.

It is apparent that there are several mechanisms that enter the phenomenon at various scales thus a multi-scale approach could be foreseen (the authors could refer for example to

  1. a)       Multi-scale modeling in nanomaterials science. Materials Science and Engineering: C, 27(5-8), 1082-1089 (2007), (2017).
  2. b)      Validity of Cauchy–Born hypothesis in multi-scale modeling of plastic deformations. International Journal of Solids and Structures, 115, 224-247 (2017).

Where lower scale methods are employed in order to extract more exact properties to enter to larger scale models.

Response 3: One of the challenges concerning bonded-particle model (BPM) simulation is it demands a high computational power due to the massive numbers of objects. This research has already implemented GPU acceleration in the BPM simulation framework. However, the calculation of a single simulation without any upscaling ranged around 2-4 hours. With the multi-scale approach, the upscaling of the simulation causes the simulation duration to increase exponentially. In particular, the numbers of bonds rise dramatically. In order to maintain a good balance between total simulation duration and good material properties estimation, upscaling of the current simulation model has not been considered in this research scope. However, it is worth studying the further upscaling of both particle and bond for more accurate properties estimation in future research.

Point 4: How the bonds are modeled on the BPM model

Response 4: The bond is modelled as a virtual cylindrical linkage between the particles. Bonds are treated as virtual objects which have no volume or mass. The bond's internal force is calculated according to strain changes; in our case, time is also considered in the bond mathematical model. Particles can interact by direct contact between particles or connection via bonds.

A description of the modelling of bonds is added to the article.

Point 5: How the frozen water is models in the BPM model. How the % saturation is taken into account in this modeling

Response 5: From section 2.3, the article mentioned the experiments are performed at a temperature well under 0 °C. With the well-ventilated condition inside the climate chamber, the possibility of unfrozen water can be minimized. All the frozen water is considered solid bonds and calculated according to the solid bond model proposed in this literature. The % saturation difference between different agglomerates is considered by reducing the total number of bonds and the bonds' cross-section area. The assumption and procedure behind the internal bond structure creation are described in section 3.2. 

Point 6: The agreement between experimental and simulation results is better as the authors mention too in the case of 100% saturation. Is this perhaps due to the fact that in that case the continuum model approach is more valid since there is no empty space as in the case of 75% saturation? And the BPM model as a continuum model performs better.

Response 6: Apart from formulating the material behaviour with the newly developed solid bond model, the agglomerate's breakage behaviour and parameter are also focused on in the investigation. Bonds are set to be destroyed according to breakage stress or breakage strain. The continuum model is brilliant for calculating deformation, rheology and fatigue problems. However, if breakage is considered in the continuum model, re-meshing and breakage criteria must be considered. The empty space 75% saturation level agglomerate is considered in the simulation model by reducing the total number of bonds and bonds' cross-section area.

Additionally, when modelling the agglomerate with different saturation levels, the material model parameter has to be alternated to compensate for the reduction in bond material. In this study, Young's modulus and Poisson's ratio are kept the same through every single set of simulations. Only the bond normal and tangential strength, creep parameter requires fine adjustment concerning different types of particles. More fine adjustment regarding material parameters is required when using the continuum model.

Point 7: Just a suggestion for the authors for future research. If they posses a significant number of experimental values they could attempt a machine learning approach to predict the values of  the Young modulus as function of the various parameters that enter the calculations (methods range from non transparent ML methods like neural networks (see for example prediction of composite microstructure stress-strain curves using convolutional neural networks. Materials & Design, 189, 108509 (2020.) to transparent methods like symbolic regression (see for example A combined clustering/symbolic regression framework for fluid property prediction. Physics of Fluids.(2022))

Response 7: Thank you very much for the ideas. It is a brilliant idea that we have investigated also. A preliminary investigation has already been done and published. Here is the publicated literature.

A)Dosta, Maksym, and Tsz Tung Chan. "Linking process-property relationships for multicomponent agglomerates using DEM-ANN-PBM coupling." Powder Technology 398 (2022): 117156.

In that literature, numerous simulations have been performed that obtained Young's modulus, breakage strain, stress and energy from simulation results. The information is then used for Artificial Neural Network (ANN) training. In that part, the ANN provides a reasonable estimation towards the material parameter prediction. However, the publicated literature has only focused on feed-forward prediction. The backpropagation with experimental results is definitely worth investigating.  

Lastly, thank you very much for your effort in reading my article. Your ideas and comment gave me great assist through the article correction and the further research direction.

Reviewer 2 Report

This work is reported by the authors from Hamburg University of Technology (TUHH), Germany. Microscale modelling of frozen particle fluid systems was done with a bonded-particle model method. A solid bond model that combines strain-dependent linear elastic behaviour with time-dependent creep behavior is proposed. The results are obtained in the DEM framework (MUSEN). 

The proposed model is a significant development in polycrystalline ice investigations. Experimental work is also done to prepare particles with different surface and mechanical properties. The experimental results validated the model and are found in a good agreement with each other. The manuscript is well written, the text require no changes. The results of this work are sound and well justified by the experimental data of this work and literature. The experiments and simulations demonstrated especial correspondence in Young’s modulus and Breakage stress. I can recommend this work for publishing as it is. 

1. What is the main question addressed by the research? The authors revealed the behaviour of strain rates, saturation levels, particle surface and so on of frozen particle fluid systems in compression experiments.
2. Do you consider the topic original or relevant in the field? Does it
address a specific gap in the field? This topic is relevant to the field, additional factors are taken into account in the proposed new solid bond model.  
3. What does it add to the subject area compared with other published
material? The experiments and simulations demonstrated especial correspondence in Young’s modulus and Breakage stress.
4. What specific improvements should the authors consider regarding the
methodology? What further controls should be considered? The methodology of theoretical and experimental studies are covered in the manuscript in detail. 
5. Are the conclusions consistent with the evidence and arguments presented
and do they address the main question posed? The conclusions are consistent with the results achieved and reported in the literature. There are no concerns about the validity of the results.
6. Are the references appropriate? All references are appropriate and cover the subject. 
7. Please include any additional comments on the tables and figures. No additional changes are required for figures and tables.

Author Response

Thank you very much for the effort on revising our article.

Reviewer 3 Report

The paper introduced novel modelling approach based on DEM and BPP to model mechanical behaviour of frozen particle fluid systems. However, there are several ‘free’ parameters in the model which were obtained by trial and error which is not shown. I would suggest some kind of sensitivity analysis to show influence of these parameters on the results  and more rigorous way to estimate them from experiments so the future researcher can use this approach more reliable. Some other comments are listed as well.  

There are several broken references in the manuscript for the figures when introduced in the text, e.g. 1st one on line 221, please correct all of these broken references.  

Provide particle size in Table 1 for each class of primary particles.

Ice Creep behaviour: how compressive Young’s modulus was obtained? Was it some fitting, if yes, please show on the plot? What was error in estimated value?

Simulation setup: Similar to the creep parameters, bond normal and tangential strength is tuned by trial and error procedure. -please provide more details on your procedure? What error was used to adjust parameters?

Small comments:

page 2 line, Poissons ratio from 0.29 to 0.32m – the ratio is dimensionless so what is m for?

Lines 95-97: for the strains put exponents in superscripts.

Lines 106-107: Ductile behaviour is mainly identified by strain-rate hardening and thermal softening, in which the activation energy is above -10 °C. – how activation energy have units of temperature? Please clarify.

Figure 2 caption is hanging, please rearrange the text so it is staying on the same page with figure.

Author Response

Thank you very much for your effort on reviewing our article, the manuscript has been adjusted according to the comments.

Round 2

Reviewer 3 Report

Authors should provide rebuttal detailing their responses to the reviewer's comments and pointing to the changes in the manuscript to speed up review process.